# Investigating clinic transfers among HIV patients considered lost to follow-up to improve understanding of the HIV care cascade: Findings from a cohort study in rural north-eastern South Africa

David Etoori[1]*, Chodziwadziwa Whiteson Kabudula[2], Alison Wringe[1], Brian Rice[1], Jenny Renju[1,3], Francesc Xavier Gomez-Olive[2], Georges Reniers[1,2]

**1** London School of Hygiene and Tropical Medicine, London, United Kingdom, **2** MRC/WITS Rural Public Health and Health Transitions Research Unit (Agincourt), School of Public Health, University of Witwatersrand, Johannesburg, South Africa, **3** Kilimanjaro Christian Medical University College, Moshi, Tanzania

* davidetoori@gmail.com

**Data Availability Statement:** Data are available on reasonable request. Data may be obtained from a

## Abstract

Investigating clinical transfers of HIV patients is important for accurate estimates of retention and informing interventions to support patients. We investigate transfers for adults reported as lost to follow-up (LTFU) from eight HIV care facilities in the Agincourt health and demographic surveillance system (HDSS), South Africa. Using linked clinic and HDSS records, outcomes of adults more than 90 days late for their last scheduled clinic visit were determined through clinic and routine tracing record reviews, HDSS data, and supplementary tracing. Factors associated with transferring to another clinic were determined through Cox regression models. Transfers were graphically and geospatially visualised. Transfers were more common for women, patients living further from the clinic, and patients with higher baseline CD4 cell counts. Transfers to clinics within the HDSS were more likely to be undocumented and were significantly more likely for women pregnant at ART initiation. Transfers outside the HDSS clustered around economic hubs. Patients transferring to health facilities within the HDSS may be shopping for better care, whereas those who transfer out of the HDSS may be migrating for work. Treatment programmes should facilitate transfer processes for patients, ensure continuity of care among those migrating, and improve tracking of undocumented transfers.

## Introduction

Although antiretroviral therapy (ART) expansion has improved life expectancy among people living with HIV (PLHIV) [1,2], several ART programmes in sub-Saharan Africa have reported increasing rates of loss to follow-up (LTFU) over time among patients initiating ART [3–5]. Increasing patient numbers have placed pressure on health systems, leading to longer waiting

third party and are not publicly available. Under our data agreements with MRC/Wits Rural Public Health and Health Transitions Research Unit (Agincourt), we are unable to share the data used for these analyses directly with others. Data are however available by request here: https://www.agincourt.co.za/?page_id=1883 and with permission of the MRC/Wits Rural Public Health and Health Transitions Research Unit (Agincourt).

**Funding:** This study was made possible with support from the Economic and Social Research Council (ES/JS00021/1 to DE), the Bill and Melinda Gates Foundation for the MeSH Consortium (OPP1120138 to BR), the Bill and Melinda Gates Foundation ALPHA grant (OPP1164897 to GR), and the MRC SHAPE UTT grant (MR/P014313/1 to AW). The funders had no role in study design, data collection and analysis, decision to publish, or preparation of the manuscript.

**Competing interests:** The authors have declared that no competing interests exist.

times and shorter counselling consultations for patients, which may increase LTFU [6,7]. Asymptomatic or less severely ill patients initiating ART may be less likely to see the merits of remaining engaged in care [7,8], and higher rates of asymptomatic disease are linked to increased LTFU among women who initiate ART for prevention of mother-to-child transmission (PMTCT) compared to the general population [9,10]. High rates of LTFU among PLHIV present multiple challenges, including increased mortality risk and HIV transmission risk [11–14].

Misclassification of patients as LTFU may lead to biased estimates of death and retention. One study reported a five-fold underestimation of death and retention rates and an overestimation of attrition due to LTFU [15]. However, as more healthy people initiate ART, less LTFU will be attributable to mortality [16,17]. "Silent" transfers majorly contribute to LTFU as sending facilities are unaware of these transfers and so they go unreported [16,18]. Decentralised ART provision in sub-Saharan Africa, reliance on paper-based registers, a lack of unique identifiers and poorly implemented electronic databases all ensure patient movements between clinics are difficult to track and document [19]. Silent transfers might also be driven by patient practices, such as "clinic shopping" as programmes expand, and ART becomes increasingly available [16,20]. A further reason may be poorly documented transfers in the wake of decentralisation efforts. One South African study reported an increase in transfers from 1.4% among patients enrolled in care between 2002–2004 to 8.9% for patients enrolled in 2009, following ART expansion [21].

Previous studies reported high mortality immediately following clinic transfers and return migration, attributed to patients with severe illness moving closer to home to be cared for in the expectation of death [22–26]. However, there is also evidence that some patients are transferring to actively seek better quality care and improved outcomes [23,27]. Historically, healthy individuals have been more likely to participate in migration [28]. Recent studies show increased participation in migration by physically healthier PLHIV [29]. Potentially increased migration among PLHIV has important implications for their continuity of care and consequently the potential transmission of HIV [30–32].

Recent research into HIV patient mobility highlights continued HIV care among patients considered LTFU after ART initiation [6,33]. This work demonstrates the increasing occurrence of patient mobility and silent transfers in HIV programmes. As such, we need to better understand how this mobility affects retention and the benefits of ART [34], and how it contributes to silent transfers and over-estimation of LTFU.

We undertook a record review and patient tracing study in rural north-eastern South Africa to understand the outcomes of patients considered LTFU, from a network of eight public sector ART facilities. We utilised routinely collected data from the Agincourt Health and Demographic Surveillance System (HDSS), linked to patient records from health facilities, to track patients' movement between health facilities. In this paper, we describe mobility of residents of the HDSS who initiated ART and who were recorded as LTFU, using geographic information system (GIS) mapping techniques and circular migration plots.

## Methods

### Setting

The Agincourt HDSS comprises 120,000 residents in 31 villages, located in Mpumalanga province, South Africa where HIV prevalence was 22.8% among adults 15 to 49 years in 2017 [35–37].

There are five primary health facilities (Belfast, Cunningmore, Justicia, Kildare and Xanthia) and three secondary community health centres (Agincourt, Bhubezi and

Thulamahashe) within the HDSS. At the time of this study, South Africa has moved to the universal test and treat strategy. In this setting, pregnant women must transfer their HIV care from integrated HIV and antenatal care facilities to general ART facilities 6–10 weeks after delivery [38]. All health facilities routinely trace patients who are late for a scheduled clinic appointment, in conjunction with two non-profit organisations, Right to Care (RtC) and Home-Based Carers (HBC) [39]. Health facility staff contact patients by phone, with a home visit organised if necessary. Patients are classified LTFU 90 days after their scheduled visit if they have not returned or do not have an outcome ascertained through tracing.

**Demographic surveillance.** Fertility, mortality and migration data are collected annually from residents, based on a comprehensive household registration system, in operation since 1992 [36,40–42]. Fieldworkers visit each household and interview the most knowledgeable adult available to obtain information on demographic events occurring since the last census [43].

**Point-of-contact interactive record linkage (PIRL).** Since 2014, HIV patient visits to ART clinics in the area are logged by fieldworkers and linked to the HDSS using Point-of-Contact Interactive Record Linkage (PIRL) [44,45]. In brief, a fieldworker conducts a short uptake interview with patients in the clinic waiting area. Consenting patients are asked for personal identifiers, used to search the HDSS database using a probabilistic algorithm. Matches are confirmed in interaction with the patients, and names of household members are used as a key attribute to adjudicate between possible matches.

## Record review and tracing study

Through the PIRL database, we identified HDSS residents aged 18 years or older, enrolled after record linkage was established in 2014 and more than 90 days late for a scheduled ART clinic appointment (LTFU) on August 15, 2017 (data extraction date).

Trained fieldworkers conducted a thorough record review on a case-by-case basis, comparing LTFU patients against (a) TIER.Net (the South African national HIV treatment electronic database [46]) (b) patient clinic files, and (c) logbooks kept by RtC and HBC. The PIRL database was reviewed for duplicate patients (different clinical records linking to the same individual in the HDSS database, which was taken as evidence of silent transfers), and residency and vital status were extracted from the HDSS database. This information was supplemented by an HBC home visit if no outcome could be ascertained and a further TIER.Net search at clinics close to patients' residences to capture further silent transfer. Data collection concluded in December 2018.

## Definitions

A patient was defined as having transferred if they had reported taking treatment at another health facility, if the health facility at which they initiated ART had communicated with and ascertained their transfer to another health facility, or if there was a record of them collecting treatment from another health facility within the HDSS. The sending facility was the facility at which a patient had initiated treatment and from where they had transferred their care. The destination facility was the facility to which the patient transferred their care. An undocumented ("silent") transfer was defined as one where the sending facility was unaware and therefore did not have this recorded as a patient outcome within its system. The sending facility distance was defined as the distance between the patient's village of residence and the sending facility. The destination facility distance was defined as the distance between the patient's village of residence and the destination facility. The transfer distance was defined as the distance between the sending and destination facility. Time out of care was defined as the number

of days between the last visit date at the sending facility and the first visit date at the destination facility.

## Statistical analyses

Patient characteristics were summarised using counts and proportions for categorical variables and medians and interquartile ranges (IQR) for continuous variables. A Pearson's chi-square test was used to test for differences in categorical variables, while a rank sum test was used for continuous variables.

A Cox regression model was used to determine the factors associated with undocumented transfer to another health facility, with all other outcomes treated as right censored. Bi-variate analyses were conducted with variables that were hypothesised to have a relationship with health facility transfer. All variables with $p<0.1$ were included in the multivariable Cox regression model. A parsimonious model was achieved using Wald tests. These analyses were performed in Stata 15 [47].

Patients' village of residence was ascertained through health facility and Agincourt HDSS records. A mid-point Global Positioning System (GPS) coordinate of the village was used to calculate the distance between patients' residence and the health facility where they initiated ART (sending facility). We obtained decimal degree coordinates for destination health facilities using the coordinates tool in Google Maps. Using ArcMap® 10.3.1 [48], the coordinates were imported to shapefiles with a WGS 1984 coordinate system. We then used ArcMap to spatially visualise the locations of these destination health facilities.

We calculated geographical distances between the GPS coordinates for sending and destination health facilities. We compared the median transfer distances and median time spent out of care using the Kruskal-Wallis test for the equality of populations. We performed a Pearson correlation comparing the sending facility distance (geographic distance between the sending facility and patient's residence) and destination facility distance.

To visualise movements between health facilities within Agincourt HDSS, we used the Circlize package in R [49,50]. A matrix of the volume of movements between health facilities was calculated for all transfers, stratified by the type of transfer (documented versus undocumented), and sex and pregnancy status at the time of ART initiation. These matrices were used to visualise flows between health facilities to identify any discernible patterns. To make the plots more legible, we ranked transfer volumes by size and visualised the largest 75% of transfers (the 25th percentile or greater).

## Ethics

Ethical approval was obtained from the London School of Hygiene and Tropical Medicine, the University of Witwatersrand and the Mpumalanga Department of Health.

## Results

### Population and transfer characteristics

Over the study period, of 3915 patients, 1017 (26.0%) met the LTFU criteria and were eligible for inclusion. Of these 1017 patients, 280 (27.5%) initiated ART for PMTCT, 737 (72.5%) met the ART initiation criteria for non-pregnant adults, 767 (75.4%) were females and 307 (30.2%) had ever migrated out of the HDSS.

Of 1017 patients LTFU, 315 (31.0%) had transferred (documented and undocumented) to another facility. The proportion of transfers differed by sex (p = 0.001), ART initiation year (p = 0.02), time on ART (p = 0.032), baseline CD4 (p = 0.027), health facility (p<0.001), and

time since last clinic appointment (p = 0.021) (Table 1). One hundred and eighty-four transfers (58.4%) outside the HDSS were to economic hubs such as Johannesburg, as well as agricultural and mining towns (Fig 1), with participants who initiated treatment in more recent years more likely to transfer their care further away (2014: 9.88 kilometres (IQR: 8.97, 18.67), 2015: 18.41 kilometres (IQR: 9.88, 231.87), 2016: 13.73 kilometres (IQR: 8.74, 105.93), 2017: 36.38 kilometres (IQR: 8.74, 358.26); p = 0.009).

**Differences between documented and undocumented transfers.** Of 315 patients transferring their care, 181 (57.5%) were documented at the sending facility, and 134 (42.5%) remained undocumented at the sending facility by the end of the study. One hundred and forty-two (78.5%) documented transfers were to health facilities outside the HDSS compared to 42 (31.3%) (p<0.001) of undocumented transfers. Fig 2 further illustrates this, showing that the largest flows of documented transfers were to facilities outside the Agincourt HDSS and that most of the largest undocumented transfers were to facilities near the sending facility (Fig 2). Documented transfers occurred a median of 1 day (IQR: 1, 45) after the last recorded visit date at the sending facility compared to undocumented transfers which occurred a median of 236 days (IQR: 20,489) after the last recorded visit date (p<0.001). Destination coordinates were available for 145 (80.1%) of documented and 124 (92.5%) of undocumented transfers. The median transfer distance for documented transfers was 36.38km (IQR: 13.73, 320.5) compared to 9.88km (IQR: 8.74, 12.37) for undocumented transfers. Fig 3 further illustrates this

**Table 1. Factors associated with undocumented transfer to another facility.**

|  | LTFU | Undocumented Transfers |  |  |  |  |
|---|---|---|---|---|---|---|
|  | 836 | 134 |  |  | n = 731 |  |
| VARIABLES | N (%) | N (%) | HR* (95% CI) | p-value | aHR* (95% CI) | p-values |
| Sex |  |  |  |  |  |  |
| Female | 620 (74.2) | 111 (82.8) | Reference | __ |  |  |
| Male | 216 (25.8) | 23 (17.2) | 0.52 (0.30, 0.88) | 0.015 |  |  |
| Age |  |  |  |  |  |  |
| 18–29 | 274 (32.8) | 58 (43.3) | 1.51 (1.25, 1.82) | <0.001 | 1.41 (1.11, 1.77) | 0.005 |
| 30–44 | 398 (47.6) | 61 (45.5) | Reference | __ | Reference | __ |
| 45–59 | 114 (13.6) | 11 (8.2) | 0.61 (0.19, 1.99) | 0.415 | 0.65 (0.20, 2.09) | 0.473 |
| 60+ | 50 (6.0) | 4 (3.0) | 0.50 (0.12, 2.06) | 0.335 | 0.60 (0.15, 2.51) | 0.488 |
| ART reason |  |  |  |  |  |  |
| Non-PMTCT | 597 (71.4) | 93 (69.4) | Reference | __ |  |  |
| PMTCT | 239 (28.6) | 41 (30.6) | 0.93 (0.57, 1.53) | 0.783 |  |  |
| ART start year |  |  |  |  |  |  |
| 2014 | 182 (21.8) | 28 (20.9) | 0.78 (0.46, 1.32) | 0.352 |  |  |
| 2015 | 330 (39.5) | 64 (47.8) | Reference | __ |  |  |
| 2016 | 288 (34.4) | 38 (28.4) | 0.84 (0.57, 1.25) | 0.398 |  |  |
| 2017 | 36 (4.3) | 4 (3.0) | 0.48 (0.16, 1.44) | 0.189 |  |  |
| Time on ART |  |  |  |  |  |  |
| ≤3 months | 276 (33.0) | 40 (29.9) | Reference | __ | Reference | __ |
| 3–6 months | 156 (18.7) | 28 (20.9) | 1.23 (0.65, 2.30) | 0.523 | 1.50 (0.65, 3.46) | 0.346 |
| 6–12 months | 178 (21.3) | 29 (21.6) | 1.25 (0.71, 2.20) | 0.439 | 1.43 (0.69, 2.97) | 0.408 |
| 12–24 months | 174 (20.8) | 31 (23.1) | 1.86 (1.31, 2.64) | 0.001 | 2.34 (1.29, 4.23) | 0.005 |
| >24 months | 52 (6.2) | 6 (4.5) | 1.84 (1.06, 3.18) | 0.029 | 2.07 (1.36, 3.13) | 0.001 |
| Baseline CD4 |  |  |  |  |  |  |
| <100 | 166 (19.9) | 24 (17.9) | 1.19 (0.55, 2.54) | 0.661 | 1.42 (0.66, 3.02) | 0.369 |

*(Continued)*

**Table 1.** (Continued)

| | LTFU | Undocumented Transfers | | | | |
|---|---|---|---|---|---|---|
| **100–199** | 158 (18.9) | 19 (14.2) | 0.96 (0.58, 1.58) | 0.881 | 1.06 (0.68, 1.67) | 0.793 |
| **200–349** | 220 (26.3) | 28 (20.9) | Reference | __ | Reference | __ |
| **350–499** | 150 (17.9) | 29 (21.6) | 1.57 (1.07, 2.28) | 0.02 | 1.59 (0.99, 2.57) | 0.055 |
| **500+** | 117 (14.0) | 25 (18.7) | 2.08 (1.20, 3.59) | 0.009 | 2.00 (1.06, 3.78) | 0.033 |
| **Missing** | 25 (3.0) | 9 (6.7) | __ | __ | __ | __ |
| **Baseline WHO stage** | | | | | | |
| **I** | 594 (71.1) | 102 (76.1) | __ | __ | | |
| **II** | 119 (14.2) | 18 (13.4) | __ | __ | | |
| **III** | 103 (12.3) | 13 (9.7) | __ | __ | | |
| **IV** | 9 (1.1) | 0 (0) | __ | __ | | |
| **Missing** | 11 (1.3) | 1 (0.8) | __ | __ | | |
| **Refill schedule** | | | | | | |
| **1 month** | 547 (65.4) | 85 (63.4) | Reference | __ | | |
| **2 months** | 199 (23.8) | 37 (27.6) | 1.20 (0.83, 1.72) | 0.328 | | |
| **3 months** | 59 (7.1) | 10 (7.5) | 1.27 (0.56, 2.92) | 0.567 | | |
| **>3 months** | 31 (3.7) | 2 (1.5) | 0.72 (0.31, 1.69) | 0.455 | | |
| **Health Facility** | | | | | | |
| **Facility A** | 22 (2.6) | 1 (0.8) | 0.30 (0.28, 0.31) | <0.001 | 0.30 (0.24, 0.37) | <0.001 |
| **Facility B** | 45 (5.4) | 13 (9.7) | 2.00 (1.93, 2.07) | <0.001 | 2.49 (2.03, 3.05) | <0.001 |
| **Facility C** | 239 (28.6) | 33 (24.6) | Reference | __ | Reference | __ |
| **Facility D** | 43 (5.1) | 6 (4.5) | 1.01 (0.95, 1.07) | 0.767 | 1.26 (1.07, 1.50) | 0.006 |
| **Facility E** | 81 (9.7) | 14 (10.4) | 1.01 (0.97, 1.05) | 0.562 | 1.66 (1.47, 1.88) | <0.001 |
| **Facility F** | 85 (10.2) | 17 (12.7) | 0.86 (0.80, 0.93) | <0.001 | 1.34 (1.04, 1.73) | 0.022 |
| **Facility G** | 57 (6.8) | 10 (7.5) | 1.18 (1.14, 1.23) | <0.001 | 1.97 (1.60, 2.42) | <0.001 |
| **Facility H** | 106 (12.7) | 16 (11.9) | 0.75 (0.70, 0.80) | <0.001 | 0.99 (0.77, 1.26) | 0.922 |
| **Facility I** | 158 (18.9) | 24 (17.9) | 0.81 (0.76, 0.86) | <0.001 | 1.00 (0.84, 1.20) | 0.960 |
| **Time since last appointment** | | | | | | |
| **≤1 year** | 442 (52.9) | 62 (46.3) | __ | __ | | |
| **1–2 years** | 294 (35.2) | 59 (44.0) | __ | __ | | |
| **>2 years** | 100 (12.0) | 13 (9.7) | __ | __ | | |
| | Median (IQR) | | | | | |
| **Sending facility distance (KM)** | 3.12 (0.65, 6.36) | 4.17 (0.65, 7.78) | 1.04 (1.00, 1.08) | 0.031 | 1.05 (1.02, 1.09) | 0.002 |
| **Ever been late for a scheduled clinic visit** | | | | | | |
| **No** | 502 (60.0) | 83 (61.9) | Reference | __ | | |
| **Yes** | 334 (40.0) | 51 (38.1) | 1.02 (0.69, 1.52) | 0.917 | | |
| **Ever migrated outside the HDSS** | | | | | | |
| **Permanent resident** | 593 (70.9) | 91 (67.9) | Reference | __ | | |
| **Migrant** | 243 (29.1) | 43 (32.1) | 1.04 (0.61, 1.77) | 0.897 | | |

* HR–Crude hazard ratio, aHR–adjusted hazard ratio.

showing the distribution of transfer distances by type of transfer, ART initiation reason, and ART initiation year (Fig 3). All documented transfers are excluded from further analyses.

**Characteristics of undocumented transfers.** After excluding 181 (17.8%) documented transfers, the remaining 836 patients were considered LTFU. Of these, undocumented transfers accounted for 134 (16.0%) of outcomes with 63 (47.0%) identified through record review, 57 (42.5%) through PIRL, and 14 (10.5%) through supplementary tracing. Women were more

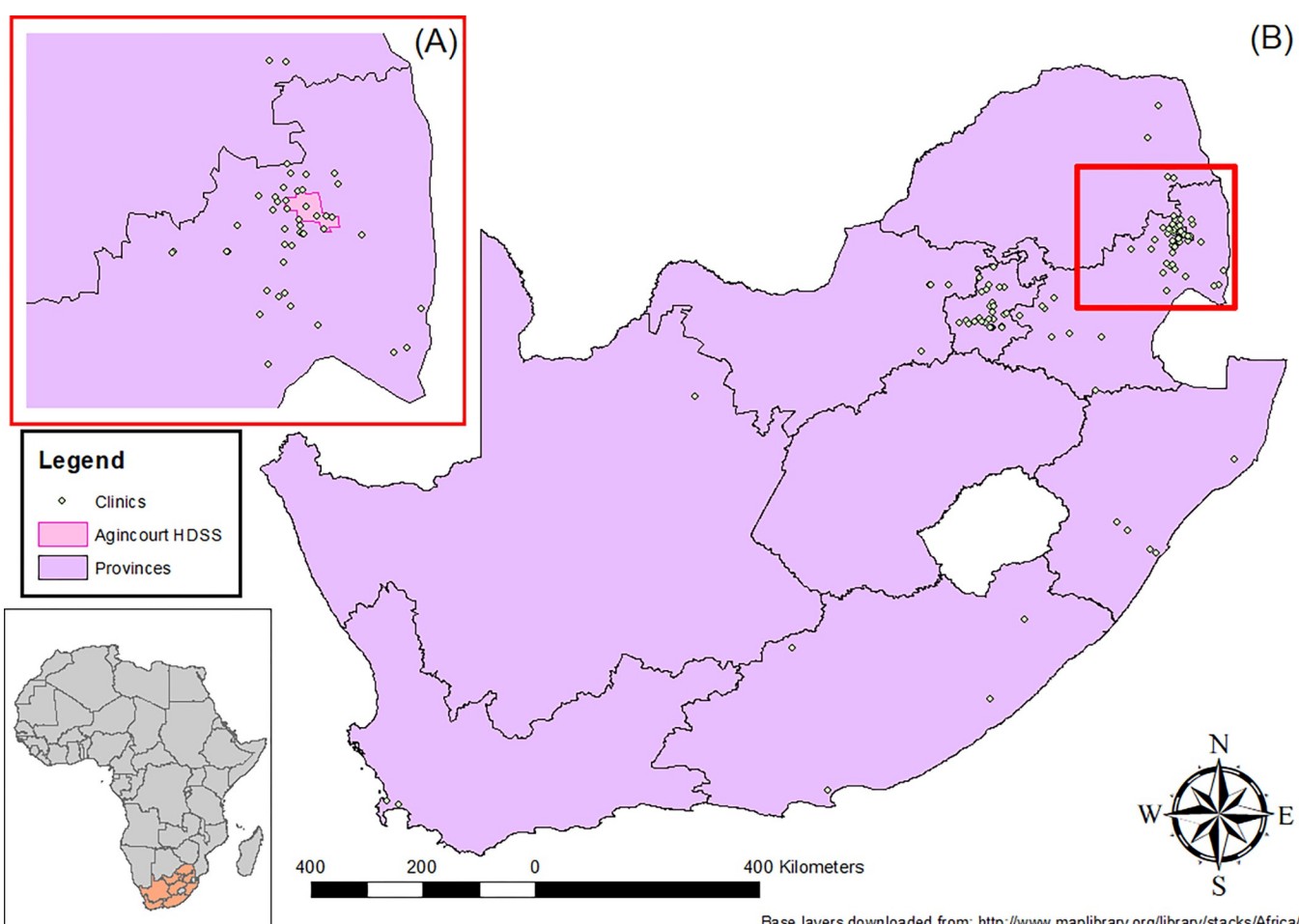

**Fig 1.** Maps of clinics attended for HIV care after transferring (A) near the HDSS and (B) within South Africa (Base layers downloaded from: http://www. maplibrary.org/library/stacks/Africa/).

likely to have an undocumented transfer (p = 0.012), with these transfers more common for non-pregnant women (18.4%, n = 381) and women who initiated ART for PMTCT (17.2%, n = 239), than among men (10.7%, n = 216) (p = 0.04). Undocumented transfers were more common for younger study participants (p = 0.01), and participants who had been LTFU for more than a year (p = 0.062) (Table 1).

Of 41 undocumented transfers for women who initiated ART for PMTCT, 9 (22.0%) were to facilities outside the HDSS, compared to 25/70 (35.7%) for non-pregnant women, and 8/23 (34.8%) for men. Fig 4 further illustrates this showing a smaller proportion of transfers out of the HDSS for women who initiated ART while pregnant and a closer balance between transfers in and out of facilities, indicating more movement between HDSS facilities (Fig 4). The median transfer distance for women was 9.88km (IQR: 8.74, 11.72), compared to 11.72km for men (IQR: 8.97, 25.70) (p = 0.0906).

Among those with undocumented transfers, men were out of care for a median of 340.5 days (IQR: 76, 452) compared to 281 days (IQR: 129, 506) for women who initiated ART for PMTCT, and 187 days (IQR: 1, 543) for non-pregnant women. Undocumented transfers to a health facility within the HDSS were out of care a median of 286 days (IQR: 129, 543) compared to 6 days (IQR: 1, 281) for those who transferred to health facilities outside the HDSS (p = 0.0004). On average undocumented transfers to health facilities within the HDSS lived

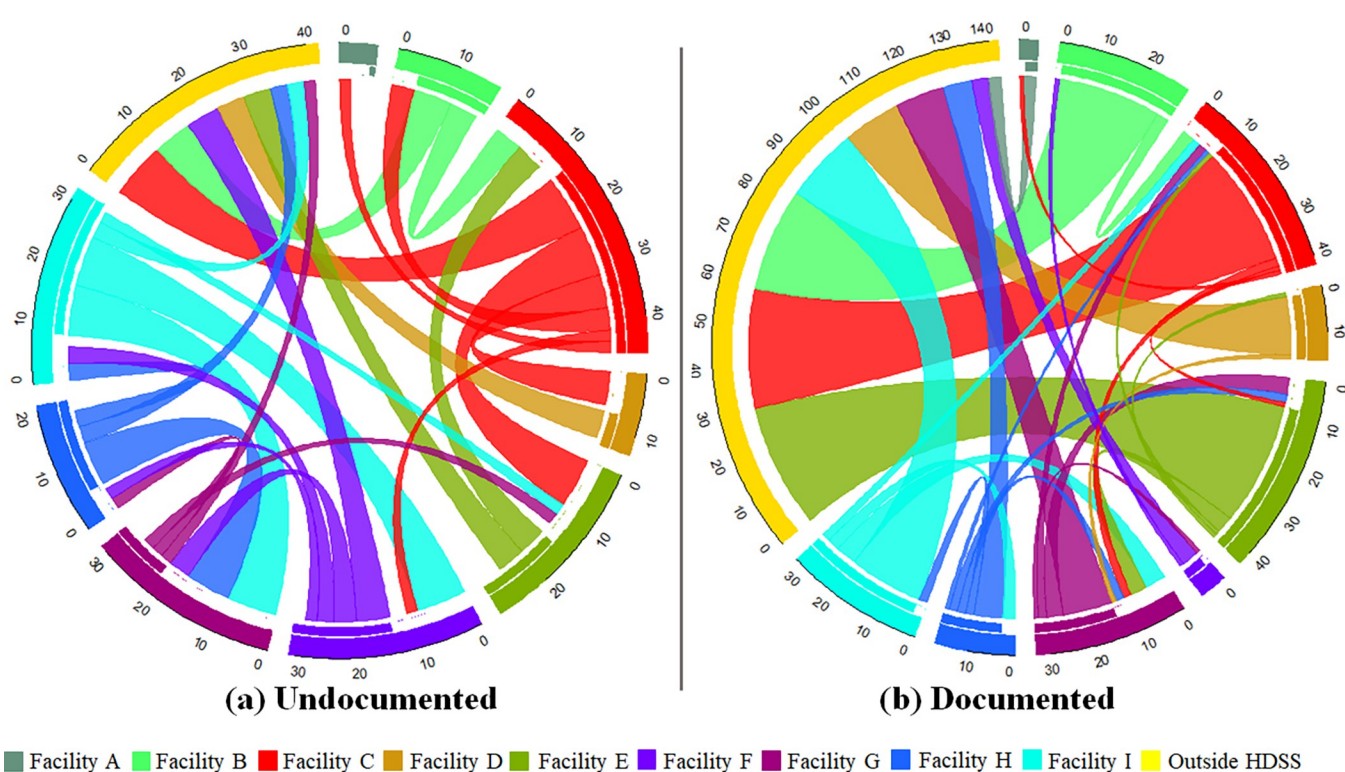

■ Facility A ■ Facility B ■ Facility C ■ Facility D ■ Facility E ■ Facility F ■ Facility G ■ Facility H ■ Facility I ■ Outside HDSS

**Fig 2. Circle plots to visualise patient transfers in the Agincourt HDSS disaggregated by type of transfer (25th percentile or greater).** The origins and destinations of transfers are represented by the colour-coded circle segments (for example Facility C is coded as red). The length of each of the outermost segment represents all the movements in and out of the particular health facility and therefore longer circle segments represent higher number of transfers to and from that facility. The inner segment represents movements out of a facility (for example because we did not investigate any facilities outside the HDSS the yellow segment representing these facilities does not have an inner segment showing transfers in but no transfers out). Segments closer together represent facilities that are geographically close to each other (within reason). As such, straight ribbons that move across the circle represent transfers further away. Facilities A and I are also the closest to the HDSS boundary. The volume of movements is indicated by the width of the ribbons and the direction of the flow is encoded by the sending facility colour (for example red ribbons represent transfers out of Facility C, also there are no yellow ribbons as we did not investigate any facilities outside the HDSS). For readability, we only visualise the largest flows (25th percentile or greater) for each transfer type.

further away from their sending health facility compared to undocumented transfers outside the HDSS (p = 0.0044). For undocumented transfers within the HDSS, there was a negative correlation between the sending facility distance compared to destination facility distance (Corr: -0.1582).

### Factors associated with undocumented transfer to another clinic

In multivariable Cox regression, younger participants (18–29 years aHR: 1.40, 95% CI 1.11–1.78), patients on ART for longer when they became LTFU (12–24 months aHR: 2.34, 1.29–4.23; >24 months aHR: 2.07, 1.36–3.13), and patients with higher baseline CD4 ($\geq$500 cells/µL aHR: 2.00, 1.06–3.78) were those most likely to have an undocumented transfer to another health facility. The hazard of an undocumented transfer increased with increasing distance between a patient's village of residence and their health facility (aHR: 1.05, 1.02–1.09) (Table 1).

### Discussion

Within a cohort of South African patients who had been categorised as LTFU, we found that 17.8% were erroneously considered to be LTFU as their transfers had been documented at their initiating facility. High rates of misclassification of LTFU most likely reflect data errors

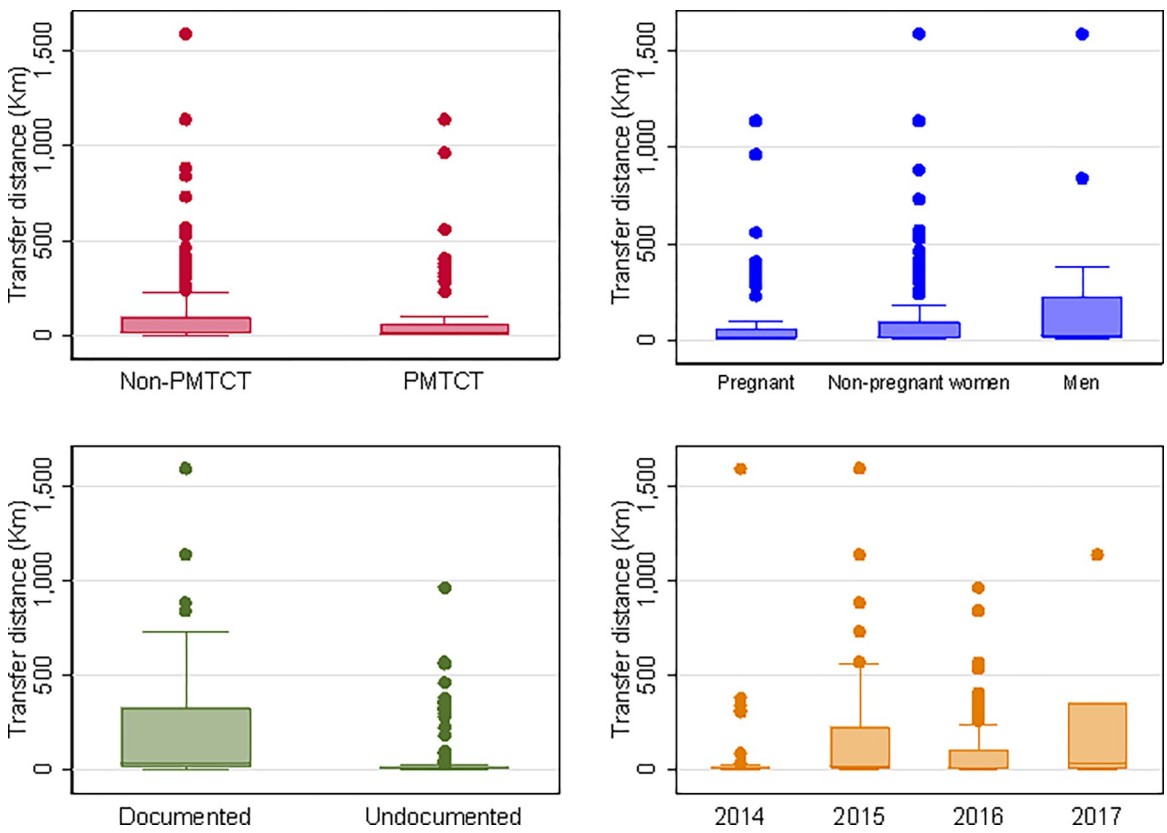

**Fig 3. Box plots of transfer distances disaggregated by ART initiation reason, pregnancy status at ART initiation, type of transfer, and year of ART initiation.**

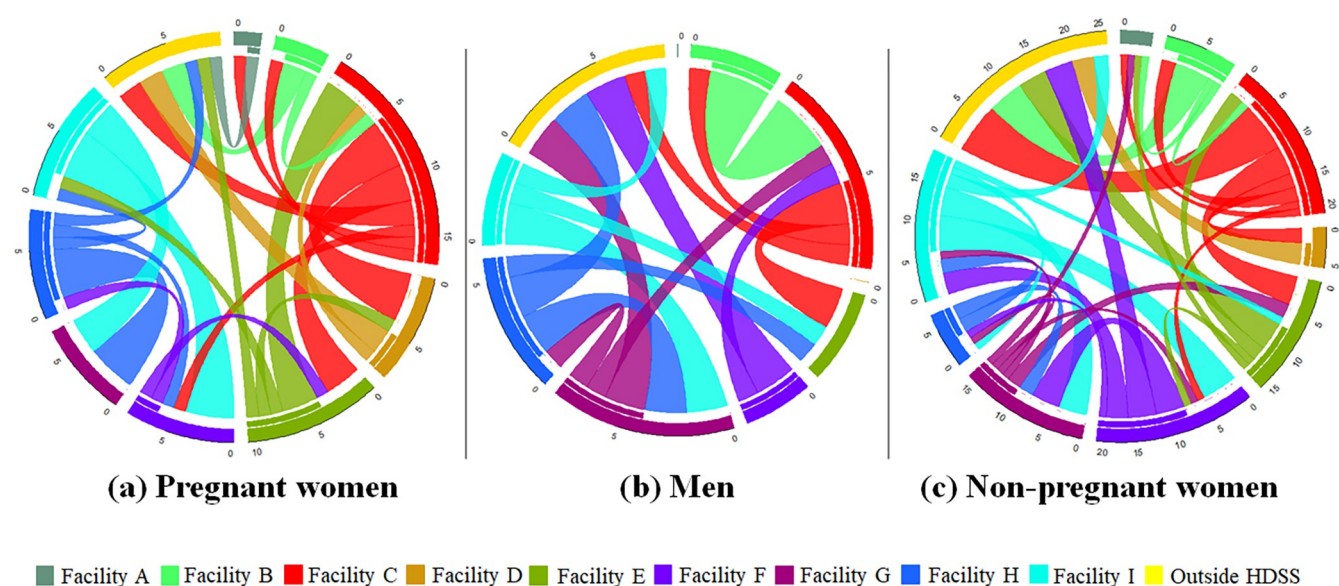

■ Facility A ■ Facility B ■ Facility C ■ Facility D ■ Facility E ■ Facility F ■ Facility G ■ Facility H ■ Facility I ■ Outside HDSS

**Fig 4. Circle plots to visualise undocumented patient transfers in the Agincourt HDSS disaggregated by sex and pregnancy status at ART initiation (25th percentile or greater).**

where transfers recorded in patient files were not expeditiously updated in TIER.Net at health facilities. This misclassification also likely reflects errors in the PIRL database where transfers were not consistently captured. As TIER.Net is still not fully adopted, fully integrating it into patient care could improve data completeness and reduce these errors. Additionally, more work needs to be done to improve recording of transfers in the PIRL database. For the remaining patients who had been LTFU (out of care for >90 days) from this original clinic, we found evidence of HIV care continuation, with 16% silently transferring their care to another facility. Our finding is similar to those reported by other studies, for example, a 2015 systematic review reported a pooled estimate of 18.6% for silent transfers among patients who had been reported as LTFU [6,16].

This study adds to extant evidence examining continued care among individuals considered LTFU after ART initiation in sub-Saharan Africa [17,51–53], and particularly transfer to other facilities [6,33]. We provide detailed comparisons of transfers by reason for ART initiation, on types of transfers and factors associated with undocumented transfer to another facility.

We found women were more likely than men to silently transfer their care, as were younger participants regardless of sex. Silent transfers among women may be linked to mobility during pregnancy and after childbirth which is high [6,13,54,55], for example as women seek assistance with childcare, and transfers for women who initiated treatment for PMTCT may reflect this. Furthermore, postpartum women in South Africa are expected to transfer their treatment to general ART clinics after delivery which might further drive these movements [33,38].

Most transfers outside the HDSS study area were to economic hubs, or agricultural and mining towns, suggesting that these transfers were due to labour migration which is common in South Africa [56]. As earlier studies suggested that PLHIV were moving back home to die [24,57], this migration is encouraging, suggesting that PLHIV feel healthy enough to get on with their lives and to participate in labour migration. This trend potentially counters challenges early in the epidemic, when HIV mortality and morbidity disproportionately affected people of working-age [58]. In our study, patients who initiated ART in later years transferred further away. As this group is also healthier with higher CD4 at initiation, this might suggest that these newer, healthier cohorts of patients may be more mobile. This is further supported by our finding that patients with higher baseline CD4 were more likely to have an undocumented transfer.

We also found longer time on ART, higher baseline CD4 and further sending facility distance to be associated with higher risk of an undocumented transfer to another facility. The negative correlation between sending facility distance and destination facility distance for transfers within the HDSS, in relation to residence, would suggest patients are moving to closer facilities. However this correlation was weak suggesting distance to facilities may not be the only factor affecting their transfers and may therefore suggest "clinic shopping" [6]. As longer time on ART was associated with undocumented transfers, this might also reflect a rational decision to move for reasons other than distance as ART scale-up has increased options. However, as patients who transferred to health facilities within the study area were more likely to be undocumented and were out of care longer than those who transferred outside the HDSS, other mechanisms such as stigma, fear of accidental disclosure, fear of healthcare worker reactions on reengaging at the same health facility, and a need to remain anonymous may influence transfers between local health facilities [6,59–63]. Longer time out of care may represent a loss of motivation to stay engaged in care and could pose higher risks for mortality, vertical and horizontal transmission [13,14,64], suggesting the need for tailored interventions to address treatment interruptions among this population [65,66]. Undocumented transfers were common, especially for women and for transfers to health facilities within the HDSS.

Undocumented transfers present several problems including the risk of amplification and transmission of HIV drug resistance when experienced patients are offered regimens with no therapeutic benefit [67]. Furthermore, this can lead to over-estimation of the number of people newly initiated on ART and ever initiated on ART, biasing ART programme indicators. Finally, these undocumented outcomes may bias estimations of retention in HIV care especially if they are misclassified as LTFU at their original clinic [16,68].

The study had some limitations. The use of a centre point for each village to represent residence may introduce some bias to our findings. The difference in completion of destination coordinates for documented and undocumented transfers may also introduce some bias as those with missing coordinates might differ from those who do. Transfers to health facilities further away from the HDSS were more likely to be documented. However, we could only check the records of health facilities within the study area and might have missed undocumented transfers to health facilities outside the HDSS, potentially introducing bias. Finally, South Africa has historically had high rates of labour migration, a legacy of apartheid [69,70], as such these results may not be generalisable to other sub-Saharan African contexts where labour migration is not as prevalent.

The use of linked demographic surveillance and clinic data is a novel approach in this setting and our findings show that it has some promise. This approach allowed us to utilise unique HDSS identifiers to follow patients between health facilities with 43% of undocumented transfers identified through this linkage. Our approach is reliant on an accurate linkage algorithm and given that some clinic records did not match to an HDSS record there is likely to be bias introduced by missed matches. Unlike fully automated record linkage, our approach minimised false matches given the interaction with the patient whose record was being linked. However, this approach is not scalable given the rarity of demographic surveillance sites which require substantial investment to set up. Furthermore, the PIRL database required substantial resources including computers for data entry and manpower to enter data, consent patients and run linkage protocols. As such, this approach may not be feasible within routine clinical follow-up. A more practical approach could be to introduce unique patient identifiers as recommended by the World Health Organisation [71]. As facility-linked data improves, so too will our ability to follow patients to monitor their long-term outcomes and ensure optimal continuation of care [33,72]. Undocumented transfers should become less prevalent with increased use of national IDs at health facilities and as TIER.Net becomes fully networked [68]. Linking clinic records to additional data sources, like the national reference laboratory database, might also improve ascertainment of transfers and has been shown to be feasible [6].

## Conclusions

We highlight the importance of ascertaining undocumented transfers as these individuals stay longer out of care, which has implications for onward transmission. Our findings suggest that an HIV diagnosis may no longer be viewed as a major impediment and people are getting on with their lives, and transfers may be linked to labour migration. As ART programmes continue to expand in sub-Saharan Africa, nationally linked treatment databases would improve the capture of transfers in highly mobile populations, to enable those who wish to transfer to do so, and to ensure continuity of care among migrants.

## Supporting information

**S1 Text. Statement on inclusivity in global health.**
(DOCX)

## Acknowledgments

The authors would like to thank all the participants in the study.

## Author Contributions

**Conceptualization:** David Etoori, Alison Wringe.

**Data curation:** Chodziwadziwa Whiteson Kabudula.

**Formal analysis:** David Etoori.

**Funding acquisition:** Alison Wringe, Brian Rice, Georges Reniers.

**Methodology:** David Etoori.

**Project administration:** David Etoori.

**Resources:** Chodziwadziwa Whiteson Kabudula, Francesc Xavier Gomez-Olive.

**Supervision:** Alison Wringe, Francesc Xavier Gomez-Olive, Georges Reniers.

**Writing – original draft:** David Etoori, Alison Wringe, Brian Rice, Jenny Renju.

**Writing – review & editing:** David Etoori, Alison Wringe, Brian Rice, Jenny Renju, Francesc Xavier Gomez-Olive, Georges Reniers.

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
