## [Decision Letter · Decision Letter 0]

25 Jan 2022

PGPH-D-21-00970

Investigating clinic transfers among HIV patients considered lost to follow-up to improve understanding of the HIV care cascade: Findings from a cohort study in rural north-eastern South Africa.

Dear Dr. Etoori,

Thank you for submitting your manuscript to PLOS Global Public Health. After careful consideration, we feel that it has merit but does not fully meet PLOS Global Public Health’s publication criteria as it currently stands. Therefore, we invite you to submit a revised version of the manuscript that addresses the points raised during the review process.

Dear authors,

Thank you for sharing your important work with PLOS Global Public Health. You will find comments from two reviewers attached. In general, please pay close attention to, and respond completely to, each of their comments. I have four additional comments:

1. Please pay particular attention to the comments from reviewer 2 that focuses on overlapping text with other publications. We noticed minor occurrences of overlapping text with the following publication: Clouse K, Vermund SH, Maskew M, Lurie MN, MacLeod 362 W, Malete G, et al. Mobility and Clinic Switching Among Postpartum Women Considered Lost to HIV Care in South Africa. J Acquir Immune Defic Syndr 1999. 2017 01;74(4):383–9. Please check all text for similarities with this, and other, publications. Ensure that you cite all your sources (including your own works), and quote or rephrase any duplicated text outside the methods section. Further consideration is dependent on these concerns being addressed.

2. Describe each figure and table in the text. For example, the circle plots are not fully described and therefore, it is difficult to understand the content, and how they fit into the findings.

3. Please provide some insight on the importance of your findings on LTFU and methodologies (using health and demographic surveillance tools to trace patients) to global health practice in general (and subsequently, this journal). This is an important opportunity as the readership of PLOS Global Public Health is not limited to HIV/AIDS practitioners and researchers. I note that the process of tracing patients is well described in the methods, but the discussion needs more work in terms of describing the important of the outcomes and transferability of these methods elsewhere.

4. Please review and update the data availability statement. 

We look forward to receiving your revised manuscript.

Kind regards,

Ruwan Ratnayake

Academic Editor

Journal Requirements:

1. Please include a complete copy of PLOS’ questionnaire on inclusivity in global research in your revised manuscript. Our policy for research in this area aims to improve transparency in the reporting of research performed outside of researchers’ own country or community. The policy applies to researchers who have travelled to a different country to conduct research, research with Indigenous populations or their lands, and research on cultural artefacts. The questionnaire can also be requested at the journal’s discretion for any other submissions, even if these conditions are not met.  Please find more information on the policy and a link to download a blank copy of the questionnaire here: https://journals.plos.org/plosone/s/best-practices-in-research-reporting. Please upload a completed version of your questionnaire as Supporting Information when you resubmit your manuscript.

2. Please amend your Financial Disclosure statement. If you did not receive any funding for this study, please simply state: “The authors received no specific funding for this work.”

3. Please update your Competing Interests statement. If you have no competing interests to declare, please state: “The authors have declared that no competing interests exist.”

4. Please provide separate figure files in .tif or .eps format only and remove any figures embedded in your manuscript file. Please ensure that all files are under our size limit of 20MB.  

For more information about how to convert your figure files please see our guidelines: 

Additional Editor Comments (if provided): none

Reviewers' comments:

Reviewer's Responses to Questions

**Comments to the Author**

1. Does this manuscript meet PLOS Global Public Health’s publication criteria? Is the manuscript technically sound, and do the data support the conclusions? The manuscript must describe methodologically and ethically rigorous research with conclusions that are appropriately drawn based on the data presented.

Reviewer #1: Yes

Reviewer #2: Yes

2. Has the statistical analysis been performed appropriately and rigorously?

Reviewer #1: Yes

Reviewer #2: Yes

3. Have the authors made all data underlying the findings in their manuscript fully available (please refer to the Data Availability Statement at the start of the manuscript PDF file)?

Reviewer #1: No

Reviewer #2: No

4. Is the manuscript presented in an intelligible fashion and written in standard English?

Reviewer #1: Yes

Reviewer #2: Yes

5. Review Comments to the Author

Reviewer #1: Comments

This submission by Etoori and colleagues aims to characterize undocumented transfers between health facilities, among patients who are initially classified as “lost to follow up” (LTFU), in the context of an antiretroviral treatment programme in rural South Africa. This is a useful addition to the literature, as there have been relatively few studies that have been able to trace patients who are LTFU in ART programmes, despite the high proportion of ART patients who are classified as LTFU in many African countries. A novel feature of this study is that it relies on existing population-level demographic surveillance activities to support this tracing. An important finding is that 18% of what is classified as LTFU is actually not LTFU but rather incorrectly processed records. Another important finding is that for those patients who are truly undocumented transfers, there is typically a long delay between when the patient is last seen at the original clinic and when they are seen at their second clinic. These delays are especially long for men, and this helps to understand the generally poorer levels of male engagement in HIV care in many sub-Saharan African ART programmes. The paper is mostly clear and well-written, although there are some gaps and some aspects that could be better explained.

Major comments

When I first read the results, I was confused by the inconsistency between the 1017 total LTFU (reported in line 190) and the 836 LTFU in Table 1. It was only when I got to the first paragraph of the discussion that I realized the reason for the inconsistency was that the former included patients who were not ‘true’ LTFU (i.e. they had been incorrectly classified as LTFU because their transfers had actually been documented). This should have been made clearer earlier on.

It would also be helpful if the authors could comment on why patients were incorrectly classified as LTFU. How general is this problem, and what can be done to limit this?

It would be good to report in the Results section how many of the undocumented transfers were identified through different means (e.g. linking to TIER versus contact through the HDSS).

It would also be good to report the overall LTFU rate, since the systematic review that the authors cite [16] shows a strong association between the LTFU rate and the proportion of LTFU cases that are undocumented transfers.

In Table 1 the authors report an “effect of time since last appointment”. It was unclear to me how they handled this in their statistical model, because in a Cox proportional hazards model the time variable is already implicit in the baseline hazard and you aren’t supposed to include it as a covariate. Perhaps I’m not understanding what the authors mean by “time since last appointment”, but it would seem as if they’re using the same variable as an outcome and as a covariate.

Minor comments

Line 71 is awkwardly phrased. Silent transfer could be explained more clearly.

Lines 85-88: I found the discussion here a bit unclear and confusing. What do you mean by “migration selection effect” and when you say “increased participation” and “increased migration”, what is that increase relative to? Relative to HIV-negative individuals, relative to HIV-positive individuals in the pre-ART era, or relative to something else?

Line 137: I don’t think the authors previously explained what TIER.Net is. It would help if this could be explained for the benefit of non-South African readers.

Figure 2: I don’t understand what the inner segment of the circle represents. The authors say in the footnote that it represents movements out of the facility, but it doesn’t equal the sum of the flows out of the segment.

Although I don’t feel strongly about it, I would consider dropping Figure 3. I don’t think it adds much to the paper, and because it doesn’t control for differences in duration of follow-up between covariate categories, it is potentially misleading. The more complete and correct analysis is already in Table 1, so presenting a simplistic analysis after you’ve already presented Table 1 is a bit distracting.

Lines 224-227: I don’t understand this sentence. Why are these “time out of care” numbers so much lower than those referred to in the previous sentence?

Lines 271-3: I don’t understand why the negative association reflects “clinic shopping”. This could be explained more clearly.

Lines 317-9: This sentence seems to be extrapolating too much from the data, and to put it as a final conclusion seems a bit weak. We don’t know for certain that these people were migrating because of employment, nor does this paper quantify the migration of people with HIV to ART services in the HDSS (i.e. migration in the opposite direction). If you’re only focusing on migration in one direction, you can’t really draw conclusions about how net migration flows have changed.

Reviewer #2: The manuscript provides a thorough analysis of documented and undocumented clinic transfers among PLWH considered lost to follow-up after ART initiation. While the concept is not novel, the authors add some new information.

My biggest problem with the manuscript is that in at least 2 places it copies nearly directly from an earlier published article -- listed as #6 on the reference list -- both in descriptions of the Methods, and also in the Discussion. For example, from the original article (published in JAIDS in 2017), "Manually searching for lost patients took many hours and is more feasible as a research exercise than as a routine clinical follow-up." The submitted manuscript states on lines 302-304: "Furthermore, searching for patient outcomes involved time consuming manual procedures and is more feasible as a research exercise than within clinical follow-up." That statement is nearly verbatim to the original statement and is too similar to be published in the manner presented, especially with no citation.

Beyond that troubling aspect, a major problem with the paper is that it does not account for required transfers during pregnancy. In South Africa, maternity care typically occurs at Midwife Obstetrics Units (MOUs) that are separate from ART facilities. After delivery, women receiving care at MOUs typically face mandatory transfer back to the ART unit for routine HIV care in the postpartum period. This was well described in citation number 33 by TK Phillips, et al, in JIAS (2018). However, the authors of the submitted manuscript do not describe the transfer requirements in place at the study sites, even though it may influence the outcome of the study, since movement among pregnant/postpartum women was found to be high.

In general, the Results are not presented very clearly. For example, lines 243-245 of the Discussion state an important point: 17.8% of LTFU were in error. But this finding is muddled in the Results. When I tried to find where the 836 number came from (line 212), I had to pull out my calculator myself. Similarly, the authors present Figures 2-5, which are fairly sophisticated from an analysis standpoint but are barely mentioned in the Results, let alone described and explained.

Lines 70-72 include a sentence that is awkward and needs revision.

Lines 187-188 note that 737 met the ART initiation criteria for non-pregnant adults. What were these criteria at the time of data collection? Define in the Methods.

Table 1: Do the 2 columns for HR and p-value represent crude and adjusted values? Something else? An important header is missing.

6. PLOS authors have the option to publish the peer review history of their article (what does this mean?). If published, this will include your full peer review and any attached files.

**Do you want your identity to be public for this peer review?** For information about this choice, including consent withdrawal, please see our Privacy Policy.

Reviewer #1: No

Reviewer #2: No

---

## [Decision Letter · Decision Letter 1]

2 May 2022

Investigating clinic transfers among HIV patients considered lost to follow-up to improve understanding of the HIV care cascade: Findings from a cohort study in rural north-eastern South Africa.

PGPH-D-21-00970R1

Dear Mr. Etoori,

We are pleased to inform you that your manuscript 'Investigating clinic transfers among HIV patients considered lost to follow-up to improve understanding of the HIV care cascade: Findings from a cohort study in rural north-eastern South Africa.' has been provisionally accepted for publication in PLOS Global Public Health.

Best regards,

Ruwan Ratnayake 

Academic Editor

Reviewer Comments (if any, and for reference):

Reviewer's Responses to Questions

**Comments to the Author**

1. If the authors have adequately addressed your comments raised in a previous round of review and you feel that this manuscript is now acceptable for publication, you may indicate that here to bypass the “Comments to the Author” section, enter your conflict of interest statement in the “Confidential to Editor” section, and submit your "Accept" recommendation.

Reviewer #1: (No Response)

2. Does this manuscript meet PLOS Global Public Health’s publication criteria? Is the manuscript technically sound, and do the data support the conclusions? The manuscript must describe methodologically and ethically rigorous research with conclusions that are appropriately drawn based on the data presented.

Reviewer #1: Yes

3. Has the statistical analysis been performed appropriately and rigorously?

Reviewer #1: Yes

4. Have the authors made all data underlying the findings in their manuscript fully available (please refer to the Data Availability Statement at the start of the manuscript PDF file)?

Reviewer #1: No

5. Is the manuscript presented in an intelligible fashion and written in standard English?

Reviewer #1: Yes

6. Review Comments to the Author

Reviewer #1: Thank you for addressing my previous comments - I am happy with the authors' responses. I have two very minor corrections to suggest (line numbering in the 'track changes' version of the manuscript):

Line 71: “contribute substantially” rather than “majorly contribute”.

Line 109: “Sought Africa had moved…”, not “Sought Africa has moved…”

7. PLOS authors have the option to publish the peer review history of their article (what does this mean?). If published, this will include your full peer review and any attached files.

**Do you want your identity to be public for this peer review?** For information about this choice, including consent withdrawal, please see our Privacy Policy.

Reviewer #1: No
